# Longitudinal monitoring of prevalence and intensity of soil-transmitted helminth infections as part of community-wide mass drug administration within the Geshiyaro project in the Bolosso Sore district, Wolaita, Ethiopia

**Toby Landeryou**[1]*, **Rosie Maddren**[1], **Santiago Rayment Gomez**[1], **Suprabhath Kalahasti**[1], **Ewnetu Firdawek Liyew**[2], **Melkie Chernet**[2], **Hussein Mohammed**[2], **Yonas Wuletaw**[2], **James Truscott**[1], **Anna E. Phillips**[1], **Alison Ower**[1], **Kathryn Forbes**[1], **Ufaysa Anjulo**[3], **Birhan Mengistu**[4], **Geremew Tasew**[3], **Mihretab Salasibew**[4], **Roy Anderson**[1]

1 London Centre for Neglected Tropical Disease Research, Department of Infectious Disease Epidemiology, Faculty of Medicine, St Marys Campus, Imperial College London, London, United Kingdom, 2 Bacterial, Parasitic and Zoonotic Diseases Research Directorate, Ethiopian Public Health Institute, Addis Ababa, Ethiopia, 3 Disease Prevention and Health Promotion Core Process, Ministry of Health, Wolaita, Ethiopia, 4 Children's Investment fund Foundation, London, United Kingdom

* tobylanderyou@gmail.com

## Abstract

Mass drug administration (MDA), targeted at school-aged children (SAC) is recommended by the World Health Organization for the control of morbidity induced by soil-transmitted helminth (STH) infection in endemic countries. However, MDA does not prevent reinfection between treatment rounds, and research suggests that only treating SAC will not be sufficient to interrupt transmission of STH. In countries with endemic infection, such as Ethiopia, the coverage, community-groups targeted, and rates of reinfection will determine how effective MDA is in suppressing transmission in the long-term. In this paper, individually-linked longitudinal data from three epidemiological STH surveys conducted between November 2018 and November 2020 in the Wolaita region of Ethiopia are analysed to determine how STH prevalence and intensity changes according to individual level treatment data collected over two rounds of MDA. This study demonstrates that while community-wide MDA successfully reduces overall infection intensity across the villages treated, the observed levels of non-compliance to treatment by individuals acts to maintain levels of parasite abundance whereby transmission interruption is not possible at to, despite reasonable levels of MDA coverage in the communities studied (ranging from 65% to 84% of the village populations). This quantifies with substantial data the often-postulated difference between coverage (accepting treatment) and compliance (swallowing of treatment), the latter impacting the former to a previously unquantified level. The paper highlights the need to focus treatment to partially treated, or never treated groups of individuals within existing community wide MDA

**Data Availability Statement:** The data presented here is epidemiological data taken as part of a survey of the Ethiopian people and therefore owned and disseminated at the discretion of Ethiopian Government. To access such data will require ethical approval from the study partner; Ethiopian Institute of Public Health https://ephi-gov.et/. To request data please contact the project lead at EPHI Dr Ewnetu Liyew Firdawek;ewnetuliyew@gmail.com and or divisional lead within EPHI Dr Geremew Tasew; getas73@yahoo.com.

**Funding:** This study was supported by the Children's Investment Fund Foundation (https://ciff.org/), via grant P74693 to RA. The funders had no role in the study design, data collection and analysis, decision to publish, or preparation of the manuscript.

**Competing interests:** The authors have declared no competing interests exist.

control activities to interrupt the transmission of STH, and to reduce the basic reproductive number, $R_0$, of the parasites to less than unity in value.

## Author summary

Soil-transmitted helminths (STH) are a group of parasites that remain a public health challenge across developing countries. The high number of global infections is ongoing despite significant financial investment in reducing the disease burden in endemic countries. In recent years WHO recommendations have shifted morbidity reduction to the possibility of transmission elimination through MDA and WaSH measures. One concern with attempting this route towards transmission interruption remains the impact that compliance and coverage has on effectiveness of community-wide treatment. The problem of STH reinfection means that a sustainable break in transmission remains difficult to achieve across an entire community. Despite repeated rounds of treatment, non-compliant individuals can shed infective eggs into the environment, essentially exposing treated individuals to become re-infected. This study highlights this problem, despite the overall effectiveness in reducing infection intensity across communities in the study area, prevalence remains at levels that would not indicate transmission interruption. This in part is due to the non-compliant individuals remaining viable reservoirs for infection that will lead to reinfection of that community before repeated round of MDA. This study indicates the importance of understanding treatment compliance and tools to combat the problem in settings where transmission interruption is being sought.

## Introduction

The soil-transmitted helminths (STHs) are a group of intestinal parasites. The dominant species in most endemic regions are *Ascaris lumbricoides* and *Ancylostoma duodenale* (roundworm), *Trichuris trichiura* (whipworm), and *Necator americanus* (hookworm). STH are thought to induce the largest morbidity burden of all neglected tropical diseases (NTDs) [1], as chronic infections are associated with growth retardation, intellectual impairment, and anaemia. There are estimated to be 1.5 billion individuals infected with at least one intestinal nematode globally, cumulatively resulting in over five million disability-adjusted life years (DALYs) [2]. The greatest burden of STH infection falls upon populations of low socioeconomic status in Southeast Asia and sub-Saharan Africa [3]. The most commonly used anti-helminthic drugs in the treatment of STH are albendazole and mebendazole [4], currently donated to control programmes by GlaxoSmithKline and Johnson & Johnson, respectively. Both are known to have few side effects [5], with sufficient efficacy to reduce prevalence and intensity of infection within endemic regions [6,7]. Repeated rounds of treatment are required since infection with STH species does not induce strong acquired immunity. Decisions on the number of annual rounds are primarily based on results from baseline infection prevalence surveys with Mass Drug Administration (MDA) typically administered in a prevalence-dependent manner, once or twice a year.

The World Health Organisation (WHO) STH treatment guidelines currently target three groups of at risk individuals; preschool children, school-aged children (SAC) and women of reproductive age [8]. Typically, SAC are the focus of treatment as the majority of donors fund integrated STH and schistosomiasis preventive chemotherapy for this age group. The WHO

guidelines also recommend the treatment of high-risk adults, which includes women of reproductive age (WRA), who can be challenging to find and treat at a community level when not attending antenatal or mother-to-child health clinics [9]. More recent STH and schistosome infection integrated control interventions have sought to interrupt transmission by expanding treatment across a broader range of age groups including adults, known as community-wide treatment. This shift in treatment focus was partially stimulated by the London Declaration in 2012, which highlighted the potential for elimination of transmission within particular settings [10]. Published research indicates that the interruption of STH and schistosome transmission is unlikely to occur when only treating SAC and WRA [11]. Transmission interruption has proved difficult to achieve primarily due to reinfection bounce back once interventions have concluded. This in part is due to the biology of soil-transmitted helminths, with eggs having the ability to survive for up to 20 years in soil substrate meaning infective potential remains within communities with continued poor WaSH practices [12,13]. There is also a growing number of studies exploring MDA acceptance within treated communities and evidence points towards untreated adult populations serving as a reservoir of infection, thus causing re-infection of treated and untreated individuals [11,14–16]. This is of particular importance when considering hookworm, as the majority of infections are harboured by adults, displayed as a rising age-prevalence distribution which plateaus in older age classes [11]. Conversely, *A. lumbricoides* and *T. trichiura* infections typically peak in younger (SAC) age groups [14].

All STH species reproduce sexually within the definitive host to shed fertile eggs into the environment. When the mean worm burden within a community drops to very low levels, there is a reduced likelihood of a male and female worm residing in a person. This should theoretically create a transmission breakpoint, whereby the worm population can no longer sustain reproduction and hence transmission, due to too few females finding a partner [17,18]. Helminth parasites are highly aggregated within human populations, with a small number of individuals harbouring the majority of worms within an infected community [19–22]. The aggregation of infection in a small number of individuals increases the likelihood that parasites successfully mate. MDA could be focused on the individuals with the highest aggregates of parasites, with the suggestion that these individuals will show a greater propensity to shed infective eggs into the community [23]. This highly focused approach has not been used at present due to the high effort and cost associated with measuring the intensity of infection within the total population, compared to treating the target population without taking a stool sample. It may be a cost-effective option when overall prevalence is very low, and drugs are no longer donated without cost to MDA programmes.

Faced with the prospect of a never-ending cycle of treatment and re-infection, a developing body of evidence has led to suggestions that endemic countries should consider expanding MDA de-worming efforts beyond the current school-based treatment system towards community-wide control [24–26]. Geshiyaro is one such project that aims to assess the impact of community-wide MDA and its feasibility in interrupting transmission with an MDA programme run by the Ministry of Health using existing infrastructure. Based in the Wolaita zone of Ethiopia, the Geshiyaro project aims to provide an evidence-based, scalable model of interventions that can interrupt transmission, eventually resulting in stopping MDA. Longitudinal sentinel site communities are monitored, measuring the annual changes in infection prevalence and intensity using standard diagnostic measures (such as Kato Katz). To evaluate treatment coverage and long-term individual treatment patterns to repeated MDA rounds, fingerprint biometric technology and study ID cards are used to register the individual uptake of MDA. Collection of individual-level treatment data allows the project to assess the impact that the proportion of individuals accepting medication at any one round of treatment (here-after referred to as MDA coverage) [27] and treatment compliance (the proportion of individuals

swallowing treatment at each round of MDA) [28,29] has on STH prevalence and infection levels within treated communities. Previous studies have indicated that the effectiveness of community-wide MDA is influenced by specific patterns of individual non-treatment [29]. This could be a random process where the probability that any individual in the target population is treated in any given round of MDA follows a positive binomial (random) distribution, or this could be systematic, where the same individuals repeatedly do not take treatment [29]. Between the limits of random acceptance of treatment, to never-treated over many rounds of MDA, a variety of possible patterns pertain for how many rounds of treatment are taken by individuals, and how past behaviour influences behaviour at the next round of MDA (the conditional probability of treatment [30]). The analyses presented in the work of Hardwick and colleagues [30] demonstrates the use of a conditional probability model in stochastic individual-based simulations by running two example forecasts for the elimination of STH transmission employing MDA within the TUMIKIA trial setting with different adherence patterns. This suggested a substantial reduction in the probability of transmission elimination (between 23–43%) when comparing the observed adherence patterns which recorded partial non-compliance, with an assumption of randomness in who gets treated at each round with defined levels of coverage.

In this paper we analyse the longitudinal STH infection data collected from a sample of individuals in the sentinel sites, as part of the Geshiyaro project in Wolaita zone, Ethiopia. The aim of this analysis is to determine the effectiveness of community-wide MDA at reducing the prevalence and intensity of STH infection across four sentinel communities, with a particular focus on three groups of individuals, fully treated, partially treated, and never treated, and their impact on community-wide infection levels.

## Methods

### Ethics statement

Data were collected in an STH epidemiological survey study whose design has been detailed in a previous publication [31]. The study received ethical approval from the Imperial College Research Ethics Committee, Imperial College London, UK and the Institutional Review Board (IRB) at the Scientific and Ethical Review Office of the Ethiopian Public Health Institute. Formal consent to receive samples from children was acquired by verbal confirmation by parent within household. All participants in the study were offered anthelmintic treatment at three rounds of annual MDA, regardless of their infection status. All treatment was directly observed.

### Parasitological mapping and selection of sentinel sites for longitudinal studies

The selection of longitudinal sentinel sites was based on the results of baseline mapping, stratifying communities by low, moderate, and high STH prevalence (baseline mapping protocol described in Mekete et al [31]). The sites were selected at random from each co-endemicity category. Sample size calculations indicated that a total of 45 sites were required for an 80% chance of detecting a true 40% reduction in STH with an intra-class correlation coefficient of 0.25, using a significance level of $\alpha = 0.05$ and $\beta = 0.8$. Longitudinal sentinel site cohort surveys using unique identifiers (biometric and/or study ID card) allows for annual monitoring of changes in infection intensity and prevalence in a sub-sample of a given population and explores the association with individual compliance to MDA. In total, 45 sentinel site communities were chosen to be sampled as part of the wider Geshiyaro project across three intervention arms; each study

arm includes MDA and WaSH interventions. The study arms and interventions that have been implemented within each designated community is described in Mekete et al [31]. In brief, the expanded WaSH and BCC measures were designed to target risk behaviour and barriers to access of sanitary water sources. Community-led total sanitation frameworks were used to build household, community, and institutional improved pit latrines. This process is used to address open defecation triggering emotions such as shame and disgust to generate collective demand for sanitation within a community. Within Ethiopia, these type of community led sanitation processes have been promoted by the national OneWASH programme [32]. Within this framework, enhanced WaSH interventions will include establishing WaSH business centres to offer products (latrine slabs, hand washing stations and soap) and services (pit digging, structural labour and maintenance of infrastructure) through subsidised sales and loans. Water infrastructure aims to provide 70% of the population through wells and taps within the first two years of the project. This will be expanded to 85% from Year 2 onwards, with the goal to achieve 82% basic improved latrines across the woreda. Behaviour change will be based on improving Health extension worker knowledge to promote good hygiene practices to minimise risk of acquiring STH. Materials will include posters, billboards and WaSH business centre catalogues focused on targeted messages to improve knowledge around STH infection. The targeted messages will include (i) handwashing with water and soap; (ii) proper household waste management; (iii) shoe wearing; (iv) safe household water handling and storage. The implementation of WaSH andBCC interventions is performed by project partner World Vision Ethiopia.

Data collection for this survey and analysis took place in one district (herein referred to as a woreda), Bolosso Sore, and from four villages (herein referred to as a kebele), Afama Garo, Korke Doge, Hajo Salata and Gido Homba. Each of these kebeles were allocated to the intervention arm designed to experience community-wide MDA, improved WaSH infrastructure and improved behaviour change communication (BCC) as described in the protocol paper [31]. In each site 150 individuals were sampled stratified by age (pre-SAC, 1–4 years; SAC, 5–14 years, 15–20 years; 21–35 years; 36+ years) with equal weighting by gender. A multi-stage cluster random sampling method was used to estimate prevalence of STH. The primary sampling unit was the kebele, which, for the baseline mapping, was selected within each woreda. Within each kebele, households were chosen using a sampling interval strategy generated using family folders at the village health post. Selected family folders were chosen according to a sampling interval number until the desired sample size was reached [33]. At each household simple random sampling was used to recruit single individuals from one of five age bands (described above) and by gender. Participants that fulfilled selection criteria (permanent village resident, provided consent, not pregnant or breastfeeding), were recruited through a random selection of households. Stool samples were collected from the participants in each survey and assessed for STH infection by trained Zonal Government Laboratory Technicians using duplicate Kato-Katz slides on two consecutive days, totalling four slides per participant [34, 35]. Egg counts were multiplied by 24 to give eggs per gram of faeces (EPG) to determine the intensity of infection [35]. The participants' stool samples were given unique identification (ID) barcodes to maintain confidentiality and to temporally link results. Stool samples were examined for eggs of *Schistosoma mansoni*, but the results are not analysed in this paper since the prevalence of infection was very low. Fig 1 gives a timeline of the data collection in these sites and when MDA took place.

## Treatment and individual registration

During the longitudinal sentinel site survey, individuals were enrolled using their census biometric/study ID card, which enabled linkage with household WASH information. Similarly,

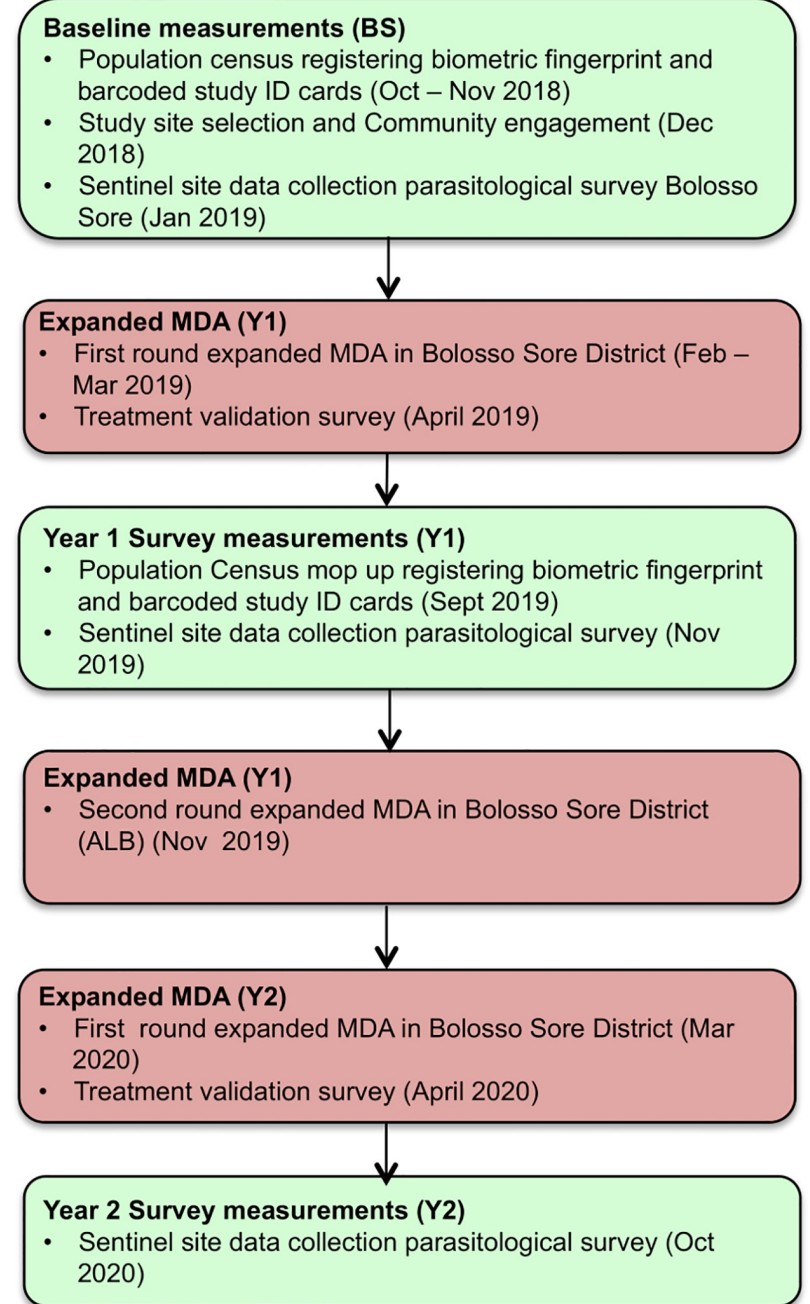

**Baseline measurements (BS)**
- Population census registering biometric fingerprint and barcoded study ID cards (Oct – Nov 2018)
- Study site selection and Community engagement (Dec 2018)
- Sentinel site data collection parasitological survey Bolosso Sore (Jan 2019)

**Expanded MDA (Y1)**
- First round expanded MDA in Bolosso Sore District (Feb – Mar 2019)
- Treatment validation survey (April 2019)

**Year 1 Survey measurements (Y1)**
- Population Census mop up registering biometric fingerprint and barcoded study ID cards (Sept 2019)
- Sentinel site data collection parasitological survey (Nov 2019)

**Expanded MDA (Y1)**
- Second round expanded MDA in Bolosso Sore District (ALB) (Nov 2019)

**Expanded MDA (Y2)**
- First round expanded MDA in Bolosso Sore District (Mar 2020)
- Treatment validation survey (April 2020)

**Year 2 Survey measurements (Y2)**
- Sentinel site data collection parasitological survey (Oct 2020)

**Fig 1. Timetable of MDA and parasitological survey across Bolosso Sore woreda.** Flow chart of time line describing the biometric registration and community engagement (BS), Expanded MDA round one and two with following parasitological survey (Y1) and midpoint parasitological survey (Y2).

during MDA individuals were registered using their biometric/study ID card, which allowed for linkage to demographic and household data as well as parasitological results. Any individuals not found to have been enrolled in the census at either the sentinel site survey or MDA, were registered as a new individual and added to the population census database. Individuals who were lost to follow-up due to geographical migration or refusal, were substituted with

new individuals added to the cohort from the least sampled age category (pre-SAC in all sites) (S1 Table).

## Statistical analysis

Statistical analysis was performed using RStudio (R version 4.1.0), this included data analysis and the generation of figures. Participants were grouped into age groups previously defined in the Geshiyaro protocol paper: pre-SAC (2–4 years old), SAC (5–14 years old), adolescents (15–20 years old), young adults (20–35 years old) and adults (36+ years old) [31]. Confidence intervals (95% two-sided) for arithmetic mean prevalence were calculated using the Clopper-Pearson method [36]. Parametric mean EPG adjusted percentiles (95% two-sided, bias-corrected and accelerated) were calculated using a bootstrapping method with 10,000 bootstrap replicates with the "*boot*" R package [37]. The WHO recommended intensity cut-offs were used to group individuals EPG into low, medium and high intensity infections [35]. We used McNemar's test to assess the differences in prevalence and Wilcoxon signed rank test to assess the differences in intensity over the surveys. Significance for longitudinal analysis of difference was set at $P \leq 0.05$. Establishing mean EPG and associated confidence limits around the mean was performed using "dbinom" R package for negative binomial distributed data. Risk ratios (RRs) were calculated by dividing the prevalence of infection in the later survey by the prevalence of infection in the earlier survey (e.g. RR = prevalence of *A. lumbricoides* at year 1 follow-up (Y1) /prevalence of *A. lumbricoides at baseline* (BS). RR confidence intervals were calculated by multiplying the standard error of the natural logarithm with the RR by the z-score and adding or subtracting the value from the log RR [36]. The significance test for the differences between RRs was derived from a formula published by Altman and Bland, 2003 [38]. Establishing overall MDA treatment coverage was performed using establishing denominator populations from baseline census as described in per the Geshiyaro protocol [31].

# Results

## Sampling enrolment

A target of 150 individuals at each of the four sentinel sites in Bolosso Sore, totalling a cohort of 600 individuals each year. Over the three years of data collection (Table 1), the Geshiyaro project has sampled 577, 621 and 573 individuals at baseline (BS), after one year (Y1), and after two years from baseline (Y2), respectively.

## Longitudinal changes in prevalence

Baseline (BS) prevalence was 32.31% for any STH, 29.7% for *A. lumbricoides*, 6.08% for *T. trichiuris* and 4.38% for Hookworm (Table 2). From BS to Y1 the prevalence of any STH

**Table 1. Sample sizes in the three surveys stratified by age and gender.**

|  | Baseline Survey | Year 1 survey | Year 2 Survey |
|---|---|---|---|
| **Age** |  |  |  |
| Pre-SAC (1-4y) | 37 | 102 | 88 |
| SAC (5-14y) | 197 | 161 | 155 |
| Adolescent (15-20y) | 58 | 79 | 64 |
| Young adult (21-35y) | 149 | 152 | 138 |
| Adult (36+) | 136 | 127 | 128 |
| **Sex** |  |  |  |
| Female | 331 | 335 | 306 |
| Male | 246 | 286 | 267 |

**Table 2. Infection prevalence recorded across STH species and yearly sentinel site surveys.** The symbol % represents the percentage positive in each group. CI = Confidence interval. EPG = eggs per gram of faeces. STH = soil-transmitted helminths.

| | Any STH | | *Ascaris lumbricoides* | | | *Tichuris trichuria* | | | Hookworm | |
|---|---|---|---|---|---|---|---|---|---|---|
| | n | % (95% CI) | n | % (95% CI) | Mean EPG (95% CI) | n | % (95% CI) | Mean EPG (95% CI) | n | % (95% CI) | Mean EPG (95% CI) |
| **Baseline survey** | 143 | 32.3 (25.12–40.41) | 134 | 29.7 (22.6–37.7) | 657 (455.88–843.22) | 35 | 6.08 (3.54–10.9) | 70.2 (14.57–129.01) | 14 | 4.38 (1.81–9.22) | 9.21 (4.46–14.05) |
| **Year 1 survey** | 170 | 36.6 (28.91–44.83) | 94 | 22.1 (15.72–29.53) | 871 (496.77–1183.78) | 48 | 9.59 (5.99–15.30) | 69.92 (17.98–121.86) | 80 | 16.4 (11.42–23.32) | 18.87 (13.02–24.72) |
| **Year 2 survey** | 127 | 31.4 (23.93–29.84) | 96 | 25.9 (18.92–34.11) | 328 (169.03–398.23) | 13 | 2.9 (1.23–6.91) | 1.1 (0.36–1.86) | 37 | 9.98 (5.56–16.27) | 5.03 (2.78–7.28) |

increased by 4.3% (percentage points; real time increase of 13.3% between BS to Y1), with the change being statistically significant ($P < 0.001$). The reduction in prevalence of *A. lumbricoides* and *T. trichuris* was statistically significant from BS to Y2 ($P < 0.05$), but hookworm showed an increase over this period. In the most recent year (Y2) of the longitudinal survey, prevalence of infection with any STH was 31.4%. compared with a 32.3% level at baseline.

There was considerable variation in trends across the four kebeles. Two kebeles had a lower prevalence of any STH in Y2 compared to BS (Afama Garo: 56.07% BS– 29.25% Y2, Korke Doge; 40.69% BS– 31.62%) and reductions in prevalence proved significant ($P < 0.001$) (Table 2 and Fig 2). Only one kebele, Afama Garo, indicated a drop in prevalence of any STH in each year of the survey. In two kebeles, Giddo Homba and Hajo Salata, prevalence of any STH increased year on year. Infection with all species peaked in Y1 before falling in Y2. An analysis of why two rounds of MDA treatment did not have a greater impact on prevalence levels is addressed in a following section where individual levels of infection are examined in relation to an individual's compliance to treatment.

Individual risk ratios for reinfection (individuals in the three time point longitudinal surveys turning from egg negative to egg positive based on the Kato Katz diagnostic) stratified by age group and gender, differed between STH species and time period over which reinfection could occur, as illustrated in Fig 3 and S2 Table. After the first round of treatment, the risk of infection reduced from BS to Y1 with any STH species ($P < 0.001$). This was largely due to a decreased risk in hookworm and *Trichuris*. There was an increased risk of infection from BS-Y1 for *A. lumbricoides* (1.34, 95% CI: 1.05–1.07) ($P < 0.05$). Surveys from Y1 –Y2 indicated increased risk of reinfection with hookworm (2.01, 95% CI: 1.38–2.91) with a significance level of $P<0.01$. Increased risk of reinfection was also demonstrated in *T. trichiura* (3.46, 95% CI: 1.9–6.32) with strong statistical significance ($P < 0.001$) (Fig 3 and S2 Table). The reason for an increase in hookworm prevalence recorded between Y1 and Y2 but not in other species, may be attributed to improved skills amongst the technical staff performing the stool examinations. Due to the time sensitive nature of slide reading for hookworm infection but not the other infections, the data may reflect improvements in the time from stool collection to examination.

## Longitudinal changes in the intensity of infection

Mean EPG increased significantly for *A. lumbricoides* ($P < 0.001$), *T. trichiura* ($P < 0.001$) and Hookworm ($P < 0.01$) from BS-Y1. However, mean EPG decreased significantly from Y1 –Y2 for infection by *T. trichiura* ($P < 0.01$) and *A. lumbricoides* ($P < 0.001$). The observed decrease in hookworm infection for Y1-Y2 was not statistically significant. Overall, from BS-Y2, the decrease was statistically significant across all species of STH (Fig 4). In all three surveys the

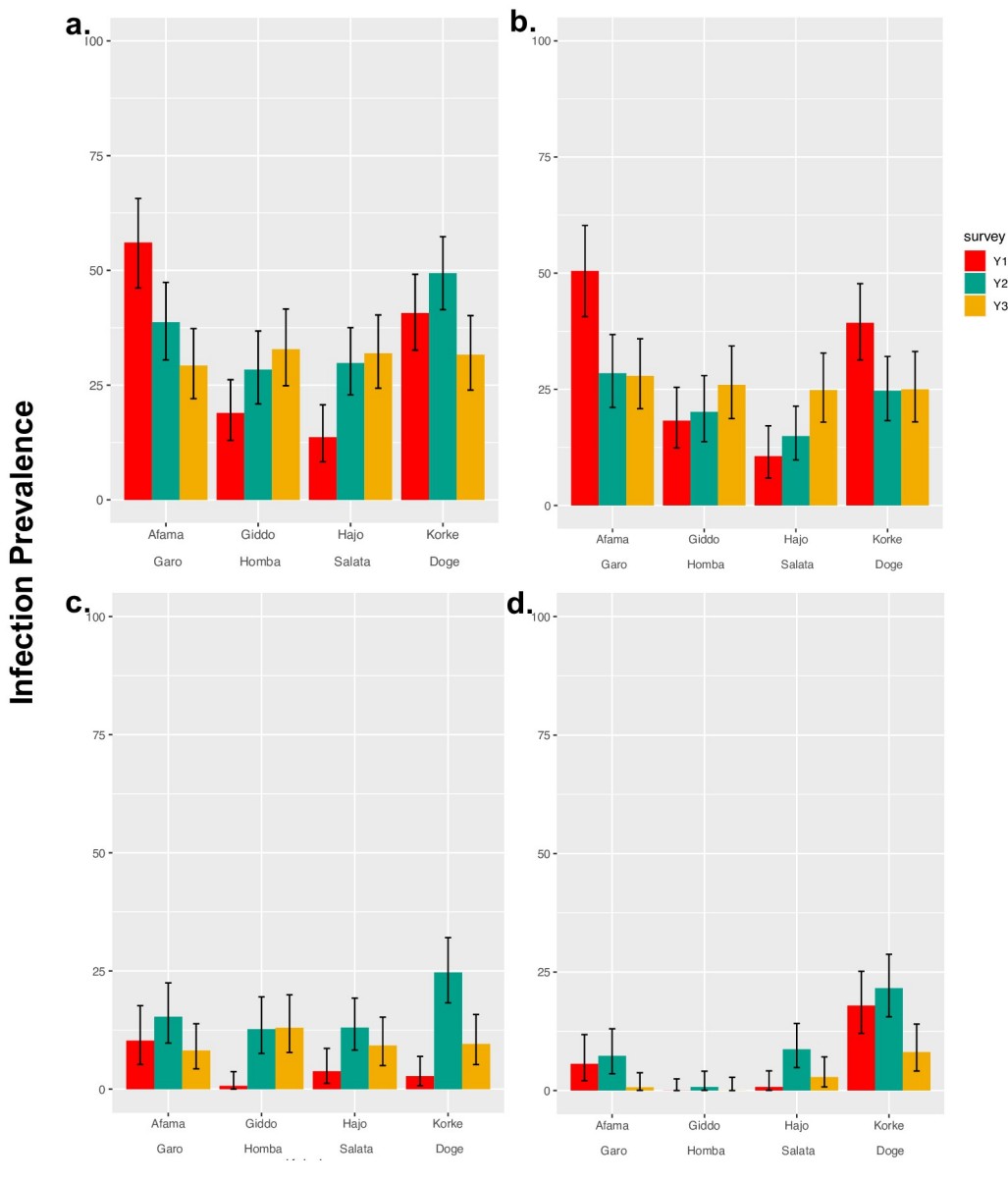

**Fig 2. Infection prevalence across cohort populations within kebeles stratified by yearly surveys.** Each plot represents species-specific prevalence of **a**. Any STH, **b**. *A. lumbricoides*, **c**. Hookworm, **d**. *T. trichiura*. Vertical Error bars represent 95% confidence intervals.

majority of infections were light intensity infections for *T.trichiura* (BS; 90%, Y1; 88%, Y2; 100%), hookworm (BS, Y1, Y2; 100%) and *A. lumbricoides* (BS; 85%, Y1; 73%, Y2; 94%). Moderate infection intensities were detected throughout all years surveyed (BS; 13%, Y1; 25%, Y2; 5%), but no heavy infections were detected in Y2 (BS; 0.6%, Y1; 1%). No moderate intensity infections were detected for *T. trichiura* after the Y1 survey (S4 Table). The categorisation of light, medium and heavy infections for all three species was as defined in by WHO [35].

When mean EPG change was stratified by age group (Fig 4 and S3 Table), mean EPG for all species increased from BS-Y1 across all age groups except infections of SAC where there was a

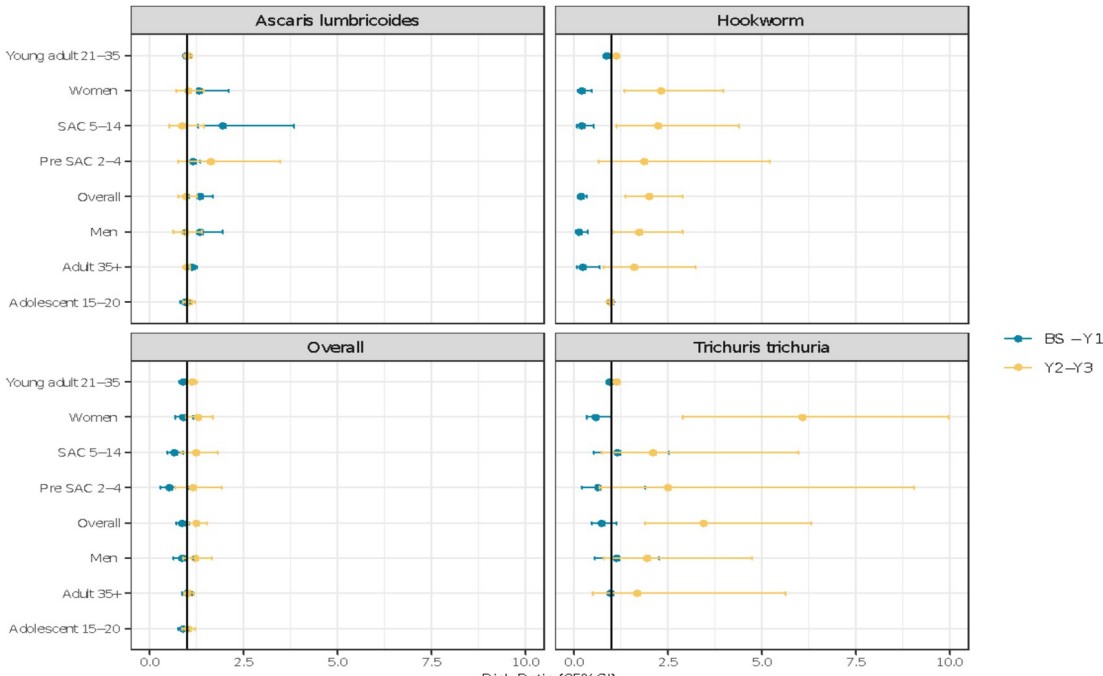

**Fig 3. Risk ratios of STH reinfection between surveys.** Blue = BS–Y1 reinfection (survey 1 to survey 2). Yellow = Y1 –Y2 reinfection (survey 2 to survey 3. Horizontal lines represent 95% confidence intervals.

decrease, which was significant ($P < 0.05$). The mean change in EPG was not homogenous across all age groups in the Y1-Y2 survey. The intensity of infection increased in SAC across both years of surveys for Hookworm.

### The impact of MDA and individual compliance to treatment

Overall MDA coverage in three of the four kebeles (Korke Doge, Afama Garo and Hajo Salata, respectively) increased from 65.8%, 81.3% and 67.1% over BS-Y1 to 79.7%, 84% and 73.1% for Y1-Y2 (S5 Table). The level of MDA coverage dropped in Giddo Homba, from 74.2% at BS-Y1: to 67.6% for Y1-Y2 (S4 Table). None of the kebeles reached the target level of 90% treatment coverage defined in the Geshiyaro protocol [31] to achieve transmission interruption. Individual level treatment data was recorded across cohort populations using the biometric and study ID card to track records of albendazole treatment. Individuals in the longitudinal cohorts were divided into three groups according to treatment compliance. These were; (1) No documented treatment over two rounds of MDA (never -"0", sample size 11); (2) Documented partaking of a single round of MDA at either Y1 or Y2 (one year–"1", sample size = 94); (3) Documented partaking in two rounds of MDA at Y1 and Y2 (two years —"2", sample size = 413).

The data records that for the period BS-Y1, 11.11% of *A. lumbricoides* and 1.02% of *T. trichiura* infections were negative based on the Kato Katz diagnostic in those individuals that received treatment in one round of MDA (Fig 5). This increased to 6.28% for *T. trichiura* and 7.04% for Hookworm infections in Y1 to Y2 but decreased in the same period to 0.9% in *A. lumbricoides*. No new individuals became infected with *A. lumbricoides* and *T. trichiura* who were uninfected at baseline assessment and received albendazole in both rounds up to Y2. Conversely, 10.14% of individuals uninfected with hookworm at BS acquired a hookworm

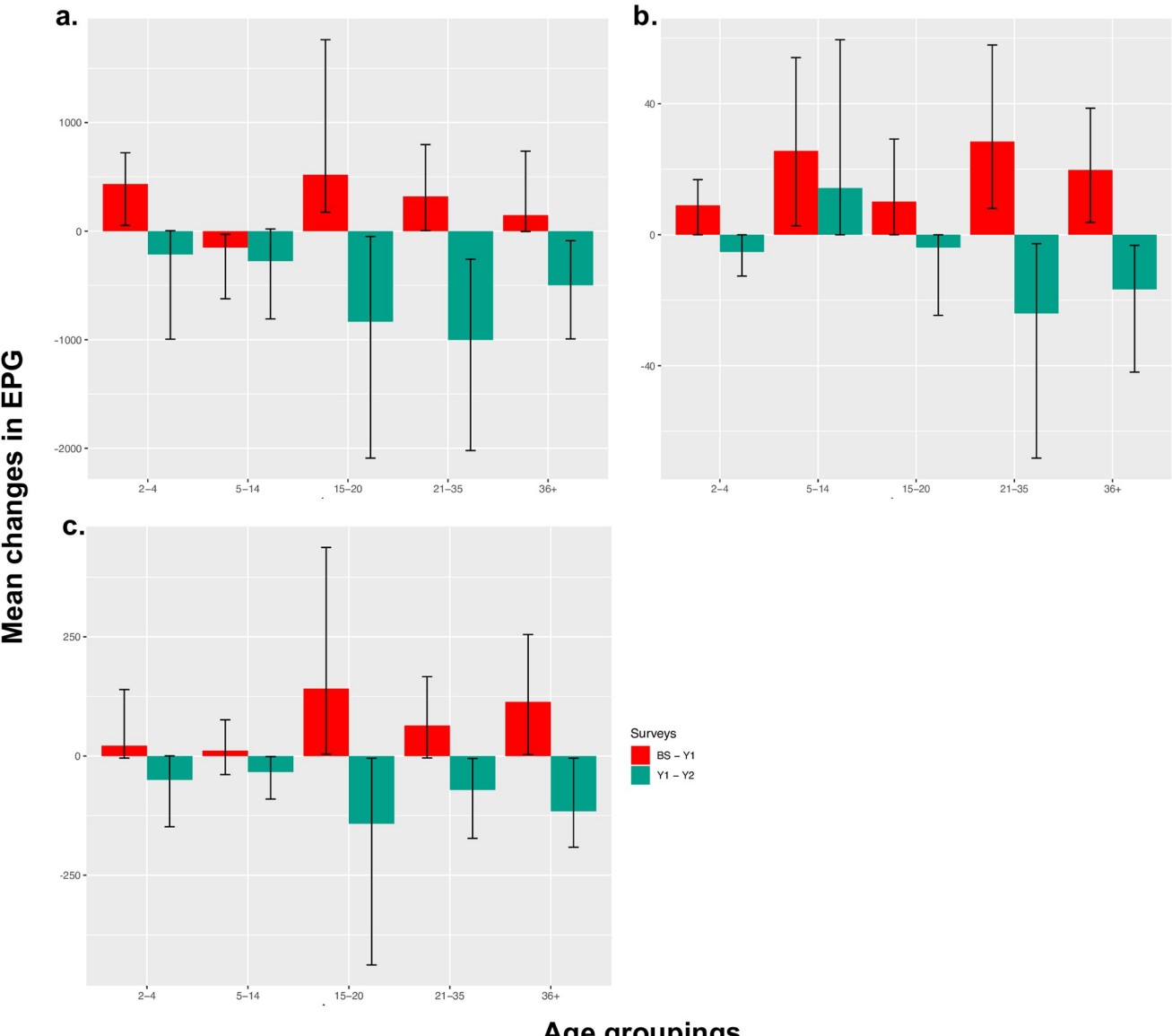

**Fig 4. Mean changes in eggs per gram of faeces (EPG) by age group.** Red bars = mean change in EPG BS–Y1. Green bars = mean change in EPG Y1 –Y2. Vertical lines represent 95% confidence intervals. Each plot represents age stratified mean epg changes across all three STH species; **a**. *A. lumbricoides*, **b**. Hookworm and **c**. *T. trichiura*.

infection between BS and Y1. This decreased to 0.48% of individuals uninfected at Y1 having an observed hookworm infection by Y2. In individuals that had received only a single year treatment, 10.53% of previously infected people did not have *A. lumbricoides* infection between BS-Y1. No one became negative for *T. trichuris* and hookworm in the same period, while 3.15% and 23.15% become newly infected with *T. trichiura* and hookworm respectively. Between Y1-Y2, each species saw a rise in negative individuals to 12.53%, 13.68% and 18.94% respectively for *A. lumbricoides*, *T. trichiura* and hookworm.

For fully non-compliant individuals (never treated) who did not partake in MDA during each of the previous years, no individual could be designated as uninfected based on the Kato Katz diagnostic method. In this no-documented treatment group, observations of acquired

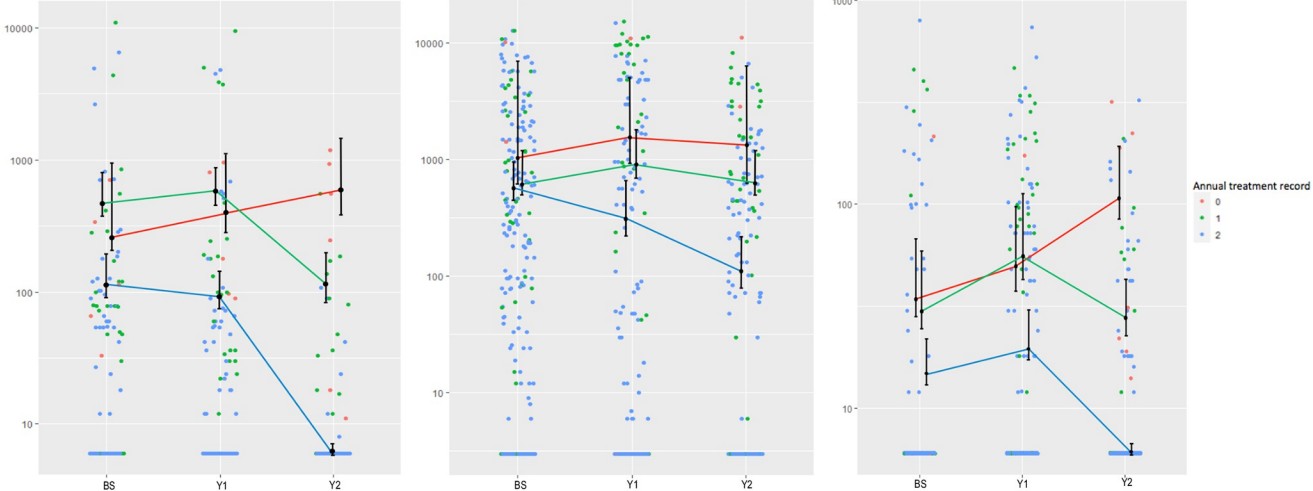

**Fig 5.** Scatter plot showing individual level intensity data of infected individuals in baseline, Year 1 and Year 2 surveys stratified by STH species; **A.** *T. trichiura*, **B.** *A. lumbricoides* and **C.** Hookworm. Each ribbon represents trend representing the mean EPG and 95% confidence limits of the mean based on the assumption of a negative binomial distribution of the epg data, of never (fully non-compliant), single (partially compliant) and two annual treatment (fully compliant) individuals.

infection (negative to positive individuals) over follow-up for all three infections are recorded in S6 Table). Age-stratified MDA acceptance data across longitudinal cohorts indicates that those individuals who were never treated through multiple rounds of MDA were all in the pre-SAC and SAC age groups. Longitudinal changes in mean EPG within each compliant age group indicated a marked difference from year to year. Across those individuals that were treated in both years of MDA, mean EPG dropped well below the level observed at the baseline survey (*T. trichiura*; 102.33, 98.22, 3.22, *A. lumbricoides*; 583.07, 331.81, 138.71, Hookworm; 12.22, 14.92, 1.37). Those individuals that had not received MDA at any round saw an increase in mean EPG from that calculated at baseline (*T. trichiura*; 422.11, 588.35, 704.2, *A. lumbricoides*; 1041.95, 1610.6, 1269.96 and Hookworm; 51.8, 62.33, 103.89) (Fig 5 and Table 3 and S6 Table). Fig 5 shows the impact on individuals of no treatment, partial compliance (i.e. one round of treatment only) and individuals who were fully compliant and took two rounds of treatment on the overall mean intensity of infection in these three groups within the sentinel site cohort studies.

## Discussion

Ethiopia launched its national deworming programme in 2015, with 100% geographical coverage. The programme has been successful in significantly reducing STH prevalence, especially

**Table 3. Percentages of individuals in all four kebeles stratified by age group and sample sizes stratified by age group, who fall into the three treatment compliance groups (never treated, treated once, and treated twice so fully treated).**

|  | Two years | One year | Never |
|---|---|---|---|
| **Pre-SAC** | 23 (5.57%) | 40 (42.55%) | 7 (63.64%) |
| **SAC** | 137 (33.17%) | 24 (25.53%) | 4 (36.36%) |
| **Adolescent** | 53 (12.83%) | 3 (3.19%) | 0 |
| **Young adult** | 106 (25.67%) | 16 (17.02%) | 0 |
| **Adult** | 94 (22.76%) | 11 (11.70%) | 0 |

in SAC-, it is still someway off achieving the very low levels of prevalence required to get close to eliminating transmission (reducing $R_0 > 1$) [39]. Whilst the long-running Ethiopian national programme targets only SAC, the Geshiyaro project seeks to evaluate the effectiveness of community-wide MDA and the addition of WASH plus BCC interventions. Interventions as a feasible and cost-effective route to the elimination of transmission. All kebeles in this sentinel site study were assigned to an arm of the programme that had all three of these improved interventions. The effectiveness of the project in terms of reducing levels of STH infection has been monitored via longitudinal studies in sentinel site communities, as detailed in Fig 5.

Previous studies measuring the longitudinal effectiveness of MDA programmes have typically focused on the prevalence of infection rather than the mean intensity. Such studies often record a substantial drop in prevalence in the first years of MDA followed by smaller reductions in subsequent years [29,40,41]. This has not been the pattern observed across sampled communities in Bolosso Sore, where each species saw a significant increase in prevalence after the first round of community-wide MDA. The overall prevalence of infection with any species of STH was reduced by only 0.9% from baseline to Year 2.

Although prevalence is used as a key metric in many STH epidemiological studies, the intensity of infection is a much more important determinant of the morbidity induced by STH infection. As such it is a better marker of the impact of interventions. The relationship between prevalence and intensity is very non-linear (as defined by the negative binomial distribution of parasite numbers per person), where prevalence changes little at high mean worm burdens but then falls rapidly at low average worm loads [17]. The majority of infected participants had low intensity of infections, with only *A. lumbricoides* recording moderate level infection intensities after the second round of MDA. The sentinel site longitudinal data indicated that the mean intensity of all STH infections was significantly reduced between baseline and Year 2 (hookworm: mean epg reduced from 9.21 to 5.03, *T. trichiura*: mean epg reduced from 6.08 to 2.9 and *A. lumbricoides*: mean epg reduced from 657.01 to 328.3).

Aggregation of STH infection within human hosts is a uniformly observed epidemiological feature of human helminth infections [17,42]. The degree of over dispersion or aggregation of infection can depend on the parasite species, host age, gender and the transmission setting (as measured by the prevalence of infection prior to the introduction of control measures) within the community [17]. Past studies have demonstrated that those with high worm burdens tend to be predisposed to this state [23]. While the causes of predisposition to heavy infection are not well understood at present, factors may include a combination of behavioural, genetic and environmental variables. The data presented in this paper shows that compliance to MDA may be an important factor-determining predisposition to high infection levels (relative to others in the community) in populations receiving preventive chemotherapy through MDA. The analyses show clearly in Fig 5 the impact of non-compliance at one or more rounds of MDA on individual and, concomitantly, the community pattern of the prevalence and intensity of infection.

Epidemiological analyses based on individual based stochastic simulations of transmission and MDA impact have shown that with a fixed level of MDA coverage, the impact of the treatment programme on overall parasite transmission within a population is dependent on individual compliance in the treated population [29]. For a high level of MDA coverage, the likelihood of reaching transmission interruption was highly dependent on assumed compliance ranging from 90% in a random non-compliance setting to 0% when there existed a moderate fraction (25% of the population) of persistent non-compliers to treatment in the community [29]. It is of utmost importance to the success of a deworming programme, particularly one aimed at interrupting transmission, that all individuals are treated and that no individuals are repeatedly missed at each round of MDA. The existence of individuals who have

never been treated within a community may occur for many reasons, including; locations where households are too remote to be regularly accessed by health extension workers, instances where individuals are not regularly at home when the health extension worker visits, or where individuals refuse treatment. Woreda or zonal/region-specific knowledge of the relative importance of these factors in defined countries, regions and communities are crucial to planning control interventions that have the highest impact in terms of reducing prevalence to a level low enough to potentially interrupt transmission, in the absence of the repeated reintroduction of infection. The capture and archiving of robust treatment compliance and infection levels is of utmost importance when considering the design of national treatment programme frameworks. This is of particular importance when seeking to measure the impact of any changes in treatment strategy against historical strategies of national control programmes at community or district level.

The effects of migration across communities and districts may present a challenge for all deworming programmes, particularly within countries experiencing a high degree of internal displacement due to conflict or climatic factors such as drought. In all deworming programmes, villages and communities are treated as an independent unit with no immigration of infectious individuals. In future analyses there must be an increased focus on employing spatial models of human movement to understand its impact on STH persistence under repeated MDA. Encouragingly, molecular epidemiological analysis of *A. lumbricoides* within communities in Kenya indicates the majority of transmission took place within the village itself [43]. However, if there is increased movement between these infective communities, the risk of transmission may increase or potentially lead to a bounce-back of infection within communities that may have eliminated infection but are subject to many visitors from areas with endemic infection. Risk of reinfection created through migration is dependent on numerous factors such as the age profile of individuals moving between villages and heterogeneity of treatment coverage/compliance between communities. Coverage (acceptance of treatment) and compliance (swallowing of treatment) at the village level is of high importance when considering migration. Within Bolosso Sore there have been no communities that have reached the MDA coverage goals (90%) of the Geshiyaro project [44], meaning a significant number of individuals are not treated within each round of MDA. Individuals that are not treated create a reservoir of infection within the community that will lead to a rapid re-emergence of infection after the cessation of MDA. This notion also applies at a larger spatial scale such as a district where low-coverage or systemically non-compliant communities can lead to re-infection of high coverage and very compliant communities via movement between villages. This increases the importance of a trained primary level health workforce to deliver MDA to a high standard of coverage across all communities when trying to achieve a large spatial-scale transmission interruption project, such as Geshiyaro. Data collected through the Geshiyaro project, and other large-scale deworming projects such as DeWorm3, will be very beneficial in providing parameters to develop spatial models to refine interventions in interrupting transmission of STH for larger spatial areas.

Data presented here highlights the continued problem in achieving levels of community-wide coverage that will reduce prevalence resulting in transmission interruption. This has proved challenging in communities with well-trained local health workers and acceptance tracking technology. While the results presented here represent a "mid-point" of the study across these kebeles, end-line surveys will show the true impact of the tools used in Geshiyaro enroute to achieving sustainable, high levels of community-wide MDA coverage and the feasibility of transmission interruption. Future research must develop new tools and new approaches to improve the impact of MDA via increasing coverage and individual compliance for the control of helminth infections. These include molecular epidemiological methods to

quantify 'who infects whom' within households and villages and between villages, tools to allow the end-user to continue to monitor individual-level treatment data, studies to enhance the identification of the factors influencing non-compliance (these may vary widely from location to location and between different cultural settings), and methods of longitudinal follow up to measure compliance that do not increase the likelihood of non-participation due to intrusive technologies such as biometrics (e.g. fingerprinting, iris scanning or facial scanning).

## Conclusion

This paper describes epidemiological analyses of a longitudinal, individual based study of the impact of non-compliance by individuals to repeated rounds of mass drug administration to the success of MDA prophylactic treatment interventions. The study was integrated within an ongoing national deworming programme run by the Ministry of Health in Ethiopia. Repeated rounds of treatment act to reduce the overall prevalence and intensity of infection. However, we have shown in this paper how the non-compliance of a small group of individuals can affect reaching the ambitious goals set for reductions in both the prevalence and intensity of infection in a wide-scale government run deworming study based on mass drug administration and enhanced WaSH activities. Increased aggregation of infection within those non-compliant individuals after two consecutive rounds of chemotherapy suggests that once prevalence has reached low levels across communities there is a need for MDA mop-up programmes to target and treat those that are non-compliant in order achieve the aim of interrupting the transmission of the soil transmitted helminths. A significant challenge is how to identify these individuals without incurring high costs from repeated parasitological surveys or the use of intrusive biometric methods to record who takes treatment in a defined population in order to target them in future rounds or mop-up activities.

## Supporting information

**S1 Table. Total number of enrolments across longitudinal sentinel site communities: Enrolments and followed up individuals across Baseline (BS), Year 1 (Y1) and Year 2 (Y2) surveys.**
(XLSX)

**S2 Table. Numerical output of risk ratio analysis.** Output disseminated by species and age group between BS-Y1 and Y1-Y2.
(XLS)

**S3 Table. Mean egg changes displayed per age group.** Mean EPG change is measured by species between BS-Y1 and Y1-Y2.
(XLSX)

**S4 Table. Numbers of individuals within each WHO classified infection intensity group.** Prevalence of the individuals indicating the defined groups of infection are grouped by species across BS, Y1 and Y2.
(XLSX)

**S5 Table. Coverage data per MDA intervention within communities.**
(XLSX)

**S6 Table. Treatment success metrics across individuals overall, treated once, both and never during the MDA.** These data are grouped via BS-Y1, BS-Y2 and by species of STH.
(XLS)

## Author Contributions

**Conceptualization:** Rosie Maddren, Alison Ower.

**Data curation:** Rosie Maddren, Santiago Rayment Gomez, Suprabhath Kalahasti, Melkie Chernet, James Truscott, Alison Ower.

**Formal analysis:** Toby Landeryou, Rosie Maddren, Santiago Rayment Gomez, James Truscott.

**Funding acquisition:** Birhan Mengistu, Mihretab Salasibew, Roy Anderson.

**Investigation:** Toby Landeryou, Ewnetu Firdawek Liyew, Melkie Chernet, Hussein Mohammed, Yonas Wuletaw, Alison Ower, Ufaysa Anjulo.

**Methodology:** Toby Landeryou, Alison Ower, Roy Anderson.

**Project administration:** Ewnetu Firdawek Liyew, Anna E. Phillips, Kathryn Forbes, Birhan Mengistu, Geremew Tasew, Mihretab Salasibew.

**Supervision:** Ewnetu Firdawek Liyew, Roy Anderson.

**Visualization:** Toby Landeryou.

**Writing – original draft:** Toby Landeryou.

**Writing – review & editing:** Toby Landeryou, Rosie Maddren, Santiago Rayment Gomez, Ewnetu Firdawek Liyew, Alison Ower, Roy Anderson.

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
