## [Decision Letter · Decision Letter 0]

1 Jun 2022

Dear Dr Landeryou,

Thank you very much for submitting your manuscript "Longitudinal monitoring of prevalence and intensity of soil-transmitted helminth infections as part of community-wide mass drug administration within the Geshiyaro project in the Bolosso Sore district, Wolaita, Ethiopia." for consideration at PLOS Neglected Tropical Diseases. As with all papers reviewed by the journal, your manuscript was reviewed by members of the editorial board and by several independent reviewers. In light of the reviews (below this email), we would like to invite the resubmission of a significantly-revised version that takes into account the reviewers' comments. 

The manuscript is of great interest to the public, since MDA programs are under much debate about their efficacy and sustainability. The manuscript was evaluated by 3 experts in the field who pointed out some aspects that need clarification and must be addressed by the authors. There are also suggestions of figures and a scheme of a timeline by reviewers #1 and #2 that will enrich the methods and data presentation. Also, if possible, the manuscript will be enriched significantly if the authors can compare the baseline prevalence of STH before the MDA program started and how MDA impacted the baseline prevalence, as suggested.

We cannot make any decision about publication until we have seen the revised manuscript and your response to the reviewers' comments. Your revised manuscript is also likely to be sent to reviewers for further evaluation.

Sincerely,

Helton C. Santiago, M.D., Ph.D

Associate Editor

Maria Periago

Deputy Editor

The manuscript is of great interest to the public, since MDA programs are under much debate about their efficacy and sustainability. The manuscript was evaluated by 3 experts in the field who pointed out some aspects that need clarification and must be addressed by the authors. There are also suggestions of figures and a scheme of a timeline by reviewers #1 and #2 that will enrich the methods and data presentation. Also, if possible, the manuscript will be enriched significantly if the authors can compare the baseline prevalence of STH before the MDA program started and how MDA impacted the baseline prevalence, as suggested.

Reviewer's Responses to Questions

**Key Review Criteria Required for Acceptance?**

**Methods**

-Are the objectives of the study clearly articulated with a clear testable hypothesis stated?

-Is the study design appropriate to address the stated objectives?

-Is the population clearly described and appropriate for the hypothesis being tested?

-Is the sample size sufficient to ensure adequate power to address the hypothesis being tested?

-Were correct statistical analysis used to support conclusions?

-Are there concerns about ethical or regulatory requirements being met?

Reviewer #1: see general comments

Reviewer #2: (No Response)

Reviewer #3: The objectives of the study are clearly stated both in the abstract and the main text and are quite clear. The study however does not indicate a hypothesis for testing in the format may be familiar with. For this study, the sample size is adequate regardless of the fact that convenient sampling was applied. 

The methods applied for the study are adequate and so enable discussions and conclusions based on the data generated. The statistical analysis is adequate but not simple and easy to understand.

**Results**

-Does the analysis presented match the analysis plan?

-Are the results clearly and completely presented?

-Are the figures (Tables, Images) of sufficient quality for clarity?

Reviewer #1: see general comments

Reviewer #2: (No Response)

Reviewer #3: For the purposes of this study, the analysis is adequate based on the study design but the statistical analysis methods are a bit difficult to understand and follow as a reader but makes good sense with the inferences made. The figures are clear.

**Conclusions**

-Are the conclusions supported by the data presented?

-Are the limitations of analysis clearly described?

-Do the authors discuss how these data can be helpful to advance our understanding of the topic under study?

-Is public health relevance addressed?

Reviewer #1: see general comments

Reviewer #2: (No Response)

Reviewer #3: The methods applied for the study are adequate and so enable discussions and conclusions based on the data generated. However, many of the research issues addressed in the discussions and therefore conclusions seem to be supported by already known programmatic facts and country experience. A stronger connection between the data and the issues needs to be provided to strengthen the evidence.

One very important suggestion for future research in the paper is the need to elucidate ‘who infects who from within and without the community and on systematic non-compliance’. It would also have been useful to compare the impact of the community-based and school-based treatment strategies since this is an ongoing discussion within the NTD community which requires more evidence in support of the community-based treatment strategy.

**Editorial and Data Presentation Modifications?**

Reviewer #1: accept

Reviewer #2: (No Response)

Reviewer #3: The manuscript is well-written in good and easy to understand language. Apart from a few typos which I am sure will be corrected. I will recommend that the paper be accepted for publication after making strong recommendations for follow-up studies on specific topics like how transmission occurs between individuals in the community and from outside the community to help address various control and elimination strategy inadequacies.

**Summary and General Comments**

Reviewer #1: General comments

The study is interesting, and among other things is showing the difficulties to reach high level of compliance even in a small area under research setting and in my opinion confirm the fact that the transmission of these infections cannot be eliminated without an improvement of sanitation to a level that impede environmental contamination with human faeces.

In my opinion the paper merit publication but need a mayor review and additional analysis:

1-Despite having mentioned in page 5 the WHO cut offs, the changes in the prevalence of the different classes of intensity over time were not clearly presented in a table or in a figure. This should be done (by parasite) because will show the changes in morbidity suffered by the targeted population. 

2- since the authors have detailed information on who has been treated and who has not, it would be interesting to see the changes in prevalence and intensity of infection of the cohort of individuals that received treatment in all the MDA.

3- in the discussion (page 9) is mentioned that WASH and BBC interventions are part of the control strategy, this additional intervention should be explained in details, especially explaining what has been done in the 4 kebeles analyzed by the authors. 

Detailed comments 

Main

4- Page 2 start of second paragraph.

The statement: “STH treatment guidelines target morbidity control through repeated school based deworming programmes” should be corrected. WHO currently recommends to target the 3 groups at risk: preschool children, school age children and women of reproductive age. The WHO guidelines should be cited (WHO 2017 https://www.who.int/publications/i/item/9789241550116 )

5- Page 2 end of second paragraph. 

“Published research indicates……. untreated individuals” the authors should mention the second factor that impede elimination of transmission that is the contamination of the environment with helminth eggs that could remain in the soil for over 10 years.

6- Page 8, first paragraph 

“None of the kebeles reached the target level of 90% treatment coverage…” this is in my opinion an indication that the 90% coverage needed according modeling studies to interrupt transmission is probably not achievable in large scale programmes (and even is small research setting). this fact should be reported in the discussion

7- Page 9, second paragraph 

“Whilst the long running ……. Interventions”

Contains one false statement: as mentioned in comment # 4 the WHO guidelines recommend to treat preschool , schools age children and WRA

Minor

8-Page 2 line 3 I would include Ancylostoma duodenale among the main species of STH

9-Page 2, line before last 

“This create a transmission breakpoint…” I would rephrase as “This should theoretically create a transmission breakpoint…” 

10- Page 2, second paragraph 

I do not consider that is correct to mention that “.. countries are considering expanding MDA….” Actually is only a group of researchers that is considering this possibility

11-Page 6, second paragraph why the authors refer to fig 1 ? this is a table

12-Page 6, third paragraph 

Better to refer to the parasites with the correct name i.e. “A. lumbricoides” and not “Ascaris”

Reviewer #2: Comments

1. Presumably this is an area that has received many rounds of historic treatment since the start of the national program in 2015 (and possibly before). Do you have any indication as what their true baseline levels are? i.e. is repeated treatment keeping infection suppressed at the current levels?

2. Methods – you mention 45 sentinel site communities included as part of the study, but only four villages in one woreda. Do the 45 sentinel sites refer to the broader Geshiyaro project, while the four villages are for this sub-study only?

3. The design and impact of WASH and BCC is not mentioned. What WASH / BCC was implemented in this woreda and is it possible to disentangle the impacts of WASH, BCC, and deworming?

4. Was there a sample size conducted for the four villages / 600 individuals? Or was it convenience sampling?

5. “From BS to Y1 the prevalence of any STH increased by 4.3%, with the change being statistically significant (P < 0.001). There was no change in prevalence, negative or positive from BS – Y1 for all species” – are these sentences contradictory?

6. “Across those individuals that were treated in both years of MDA, mean EPG dropped well below the level observed at the baseline survey (T. trichiura; 102.33 , 9 98.22, 3.22, A. lumbricoides; 583.07, 331.81, 138.71, Hookworm; 12.22, 14.92, 1.37).”It’s good to see treatment works when it’s received. The biggest challenges here (and elsewhere) is that treatment coverage is not high enough, right? Given the experience that no communities achieved the ambitious 90% coverage target even given the extra focus of the study, and that the reductions in infection were disappointing, should we reassess the feasibility of transmission interruption?

7. I’m confused by this line: “The data presented in this paper shows that compliance to MDA may be an important factor determining predisposition to high infection levels (relative to others in the community) in populations receiving preventive chemotherapy through MDA”. This is just saying that MDA reduces infection, right? Not that there’s some other mechanism. It’s the predisposition bit that’s confusing me.

8. Figure 1 refers to a timeline in the text but is a table of characteristics (I think a timeline would be useful).

Minor / Editorial

1. Introduction – is it correct to say that ALB/MEB have generally high efficacy against all species of STH?

2. Introduction – I’d recommend changing ‘eradication of transmission’ to ‘elimination of transmission’

3. It’s a minor point but decisions on treatment frequency (annual / biannual) are rarely taken on the “intrinsic transmission potential in a defined community (the magnitude of the basic reproductive number, R0)”. It’s more often taken on baseline prevalence of infection plus a range of logistical factors.

4. Conclusion: “The study was integrated within and ongoing a national…” should probably be “The study was integrated within an ongoing national…”

5. The 4.3% increase in STH prevalence from BL to Y1 – that’s percentage point rather than percentage, is that correct? So the percent increase is more like 13.3%

6. Figures – All figure titles should be able to be read standalone. Figures 3 and 4 can’t be currently.

Reviewer #3: This is an interesting article on the control of soil-transmitted helminthiasis in Ethiopia. Of particular interest to me is the fact that the survey was embedded in the program and not an isolated research activity. This provides the opportunity to collect data that may not have to be tested under separate programmatic conditions as a way of validation. The abstract is also succinct and clear. The results generated should further contribute to the development of WHO guidelines by strengthening the evidence for the control and elimination of soil-transmitted helminthiasis and will be very useful to technical advisory groups including the WHO technical advisory group on soil-transmitted helminthiasis. The results, discussions, and conclusions provide critical questions that should inform future research in this field.

PLOS authors have the option to publish the peer review history of their article (what does this mean?). If published, this will include your full peer review and any attached files.

Reviewer #1: No

Reviewer #2: Yes: Michael French

Reviewer #3: No
---

## [Decision Letter · Decision Letter 1]

19 Aug 2022

Dear Dr Landeryou,

We are pleased to inform you that your manuscript 'Longitudinal monitoring of prevalence and intensity of soil-transmitted helminth infections as part of community-wide mass drug administration within the Geshiyaro project in the Bolosso Sore district, Wolaita, Ethiopia.' has been provisionally accepted for publication in PLOS Neglected Tropical Diseases.

Best regards,

Helton C. Santiago, M.D., Ph.D

Academic Editor

Maria Periago

Section Editor

Reviewer's Responses to Questions

**Key Review Criteria Required for Acceptance?**

**Methods**

-Are the objectives of the study clearly articulated with a clear testable hypothesis stated?

-Is the study design appropriate to address the stated objectives?

-Is the population clearly described and appropriate for the hypothesis being tested?

-Is the sample size sufficient to ensure adequate power to address the hypothesis being tested?

-Were correct statistical analysis used to support conclusions?

-Are there concerns about ethical or regulatory requirements being met?

Reviewer #2: (No Response)

Reviewer #3: Longitudinal monitoring of prevalence and intensity of soil-transmitted helminth infections as part of community-wide mass drug administration within the Geshiyaro project in the Bolosso Sore district, Wolaita, Ethiopia

This research paper is illuminating and informative providing data and evidence in support of the potential global elimination strategy for soil-transmitted helminths infection in endemic populations. The choice of Ethiopia is also strategic considering the levels of endemicity in Ethiopia and also the phase of the control programme.

The study results will be of great value to the WHO’s soil-transmitted helminths strategy considering the current drive to transition from control to elimination of the global programme. All this requires adequate evidence to inform this shift which makes the contribution made available by this work and in the paper significant.

The paper is very comprehensive providing adequate background data to raise and clearly justifies the implied research questions and also the objectives of the study which demonstrate that while community-wide MDA successfully reduces overall infection intensity across the villages, the observed levels of non-compliance to treatment by individuals reduces the potential for transmission interruption despite adequate levels of MDA coverage in communities. The bottlenecks associated with low coverage, non-compliance, and non-treatment among sub-populations of endemic communities requiring attention in order to suppress infectivity within communities have been well elucidated.

**Results**

-Does the analysis presented match the analysis plan?

-Are the results clearly and completely presented?

-Are the figures (Tables, Images) of sufficient quality for clarity?

Reviewer #2: (No Response)

Reviewer #3: The method is sound and robust leading to the results obtained, discussions, and conclusions and the challenges of the study are also identified. The method is appropriate and adequate for the survey design and is well described. The statistical analysis is adequate as the study was mainly qualitative and did not require a very robust statistical analysis. The paper clearly indicates that ethical approval was acquired. The manuscript makes very good conclusions in regards to the potential transition of STH programmes from control to elimination and where the programmes need to focus on with their treatment strategies.

**Conclusions**

-Are the conclusions supported by the data presented?

-Are the limitations of analysis clearly described?

-Do the authors discuss how these data can be helpful to advance our understanding of the topic under study?

-Is public health relevance addressed?

Reviewer #2: (No Response)

Reviewer #3: Very clear conclusions supported by the method, data, and data analysis are made together with recommendations.

**Editorial and Data Presentation Modifications?**

Reviewer #2: (No Response)

Reviewer #3: There are also a few grammatical errors that should be identified and corrected such as on line 31 on the abstract page.

This is an exceptional piece of research work and abstract and I recommend it for publication after a thorough editorial work.

**Summary and General Comments**

Reviewer #2: No further comments from me. Thanks for addressing the comments and questions from the first round of review.

Reviewer #3: This research paper is illuminating and informative providing data and evidence in support of the potential global elimination strategy for soil-transmitted helminths infection in endemic populations. The choice of Ethiopia is also strategic considering the levels of endemicity in Ethiopia and also the phase of the control programme. This is an exceptional piece of research work and abstract and I recommend it for publication after a thorough editorial work.

PLOS authors have the option to publish the peer review history of their article (what does this mean?). If published, this will include your full peer review and any attached files.

Reviewer #2: **Yes: **Michael French

Reviewer #3: No

---

## [Editor Report · Acceptance letter]

12 Sep 2022

Dear Dr Landeryou,

We are delighted to inform you that your manuscript, "Longitudinal monitoring of prevalence and intensity of soil-transmitted helminth infections as part of community-wide mass drug administration within the Geshiyaro project in the Bolosso Sore district, Wolaita, Ethiopia.," has been formally accepted for publication in PLOS Neglected Tropical Diseases.

Best regards,

Shaden Kamhawi

co-Editor-in-Chief

Paul Brindley

co-Editor-in-Chief
